# Regulative synthesis of capsular polysaccharides in the pathogenesis of *Streptococcus suis*

**Xingye Wang, Jie Wang, Ning Li, Xin Fan, Beinan Wang***

Key Laboratory of Pathogenic Microbiology and Immunology, Institute of Microbiology, Chinese Academy of Sciences, Beijing, China

## eLife Assessment

This **useful** study uses a model of Streptococcus suis (a pig pathogen) infection in mice using an intranasal route, the natural route of infection ignored in most of the literature. The study aims to understand how capsular polysaccharides (CPS) contribute to neuropathology and virulence. The findings suggest that the olfactory route may lead to meningitis before bacteremia occurs and that CPS down-regulation may play a role in this process. However, the study remains **incomplete** as presented.

*For correspondence:
wangbn@im.ac.cn

Competing interest: The authors declare that no competing interests exist.

**Abstract** *Streptococcus suis* (*S. suis*) is an important zoonotic pathogen causing substantial economic losses in the swine industry. *S. suis* serotype 2 (SS2) is often isolated from the diseased. *S. suis* expresses capsular polysaccharide (CPS), a virulence factor crucial for their survival in the blood. However, the role of CPS in the pathogenesis of *S. suis* is incomplete. Here, we showed that thin CPS or no CPS was associated with efficient binding of an SS2 strain, 05ZYH33, to respiratory epithelial cells, while thick CPS increased resistance of 05ZYH33 to blood clearance. In a mouse infection model, 05ZYH33 was detected in the nasal-associated lymphoid tissue (NALT) and cerebrospinal fluid (CSF) as early as 30 min after intranasal inoculation without bacteremia. Histological analysis revealed that 05ZYH33 in the nasal cavity invaded the olfactory epithelium, resulting in early brain inflammation. Transmission electron microscopy showed that 05ZYH33 isolated from NALT and CSF at early infection time had a thin layer of CPS, and those detected in the blood 5 hr post-inoculation showed a much thicker CPS. In addition, adoptive transfer of anti-CPS restricted 05ZYH33 in the blood but not in NALT or CSF. However, an antiserum directed to multiple non-CPS virulence factors (anti-V5) efficiently inhibited 05ZYH33 in NALT, CSF, and blood. Thus, 05ZYH33 colonizes NALT more efficiently without CPS and subsequently invades the meninges through the olfactory nerve system. These findings provide valuable information for the treatment of *S. suis* infection and the development of vaccines across serotypes of *S. suis* by targeting CPS-independent immunity.

## Introduction

*Streptococcus suis* is a zoonotic pathogen in the porcine industry. Thirty-three serotypes of *S. suis* have been described to date based on the capsular polysaccharides (CPS). The *S. suis* serotype most frequently isolated from diseased swine and humans is serotype 2 (SS2). *S. suis* is commonly found in nasal-associated lymphoid tissue (NALT) of asymptomatic piglets. Under some unknown conditions, *S. suis* can reach and survive in the blood and invade the brain to cause meningitis and multi-organ dissemination. Infection of the central nervous system (CNS) is a cause of mortality in piglets with the

disease. CPS is essential for the survival of *S. suis* in the bloodstream and is recognized as an important virulence factor of this bacterial pathogen. However, the role of CPS in the pathogenesis of S. *suis* and the function of CPS-induced immunity in SS2 protection is controversial (*Lun et al., 2007*). The opsonizing antibodies targeting CPS are protective (*Smith et al., 1999*; *Fittipaldi et al., 2012*) but only partially protect against infection (*Charland et al., 1997*). Furthermore, a CPS-glycoconjugated vaccine protected parenterally challenged piglets (*Goyette-Desjardins et al., 2019*), but protection by anti-CPS has not been demonstrated in animals challenged through the intranasal route, which is a common natural infection route of *S. suis*.

Synthesis of CPS is enhanced in the bloodstream and reduced in the CNS (*Wu et al., 2014*; *Willenborg et al., 2011*). In vitro experiments revealed that the presence of CPS attenuated 05ZYH33 invasion of epithelial cells (*Schwerk et al., 2012*; *Tenenbaum et al., 2009*; *Benga et al., 2004*) and that unencapsulated mutants displayed increased rates of invasion of the blood–cerebrospinal fluid barrier (BCSFB) (*Schwerk et al., 2012*; *Tenenbaum et al., 2009*). These observations suggest that *S. suis* regulates CPS synthesis to fit hosts. However, despite the proposal of this hypothesis more than two decades ago, there is still no direct evidence of such encapsulation modulation for *S. suis* (*Gottschalk and Segura, 2000*).

While CPS has been known for years and considered necessary for infection yet probably not sufficient, other surface-associated and secreted proteins of *S. suis* have received attention during the last years for their important virulent roles in the pathogenesis and as vaccine candidates to raise effective immune response (*Fittipaldi et al., 2012*). To cause disease, *S. suis* must colonize the respiratory mucosa, breach epithelial barriers, survive in the bloodstream, spread to different organs, and exaggerate inflammation. CPS and these virulent proteins contribute to the pathogenesis of the infection at specific steps.

Two pathways have been exploited by meningeal pathogens to access the CNS: penetrating the blood–brain barrier (BBB) or the BCSFB through the blood or directly accessing the CNS via the olfactory or trigeminal nerves (*Dando et al., 2014*). Some human meningeal pathogens, such as *Streptococcus pneumoniae* and *Neisseria meningitidis*, can directly transmigrate from the nasal mucosa into the CNS through the olfactory nerve (*Dando et al., 2014*; *van Ginkel et al., 2003*; *Sjölinder and*

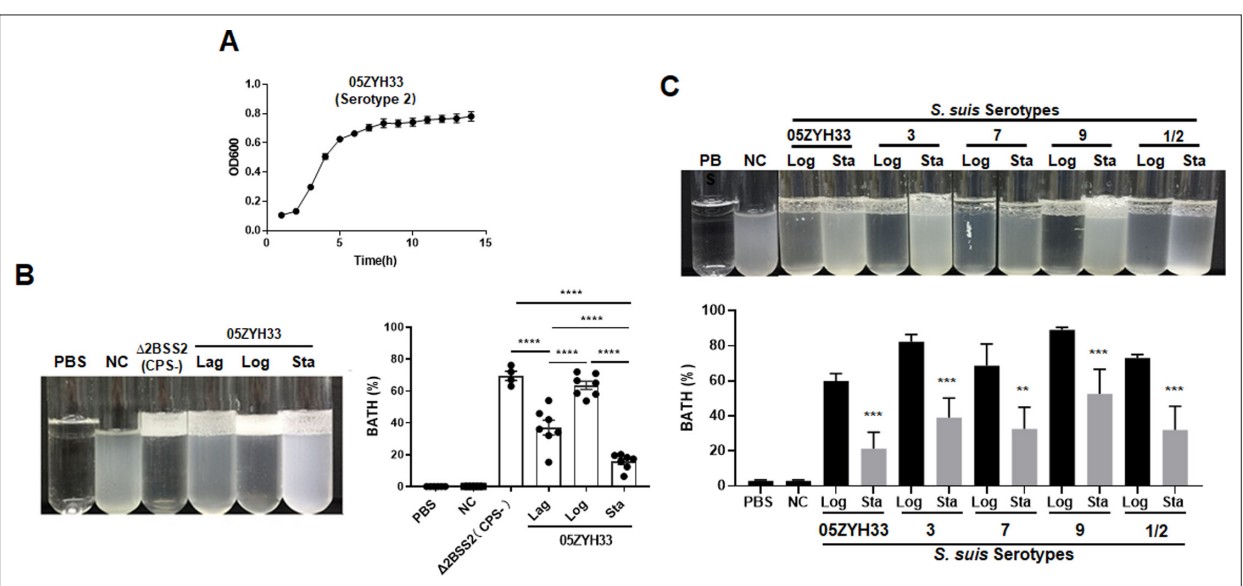

**Figure 1.** Monitoring the capsular polysaccharide (CPS) content of *S. suis* serotypes in culture. (**A**) The growth curve $10^8$ of 05ZYH33 (serotype 2) in tryptic soy broth (TSB)-FBS culture was assessed by measuring the optical density at 600 nm (OD600) at specified time points. Data present the mean ± SD of three independent experiments (n=10). (**B**) The hydrophobicity of 05ZYH33 during the lag, log, and stationary (Sta) phases and the CPS-deficient strain Δ2BSS2 (log phase) was measured by bacterial adhesion to hydrocarbon (BATH) assays (n=4–7). (**C**) The hydrophobicity of indicated serotype strains in the log and stationary phases was measured by BATH assays. A representative photo of the BATH assay (upper panel) and BATH values (lower panel) (n=3–6). Δ2BSS2 refers to an isogenic strain lacking CPS; nasal cavity (NC) represents a 05ZYH33 suspension without hydrocarbon. A one-way analysis of variance (ANOVA) with Tukey's post hoc test was utilized in B, and an unpaired, two-tailed Student's t-test was employed while in C. **, p<0.01. ***, p<0.001. ****, p<0.0001.

Jonsson, 2010). Similar to these meningeal pathogens, the predominant route of *S. suis* infection in pigs is through the upper respiratory tract (URT), which may provide the opportunity for *S. suis* to transmigrate directly to the CNS from the nasal cavity (NC).

Studies on *S. suis* commonly use parenteral infection models that bypass colonization of the URT; thus, this critical step of *S. suis* pathogenesis is largely unexplored. In the present study, mice were infected intranasally (i.n.) with 05ZYH33, and the infection process was monitored. We show that 05ZYH33 regulated the synthesis of CPS at different stages of infection and that 05ZYH33 migrated to the CNS from the NC independent of CPS before it entered the blood. Moreover, we revealed that immunity against non-CPS virulence factors of *S. suis* efficiently prevented nasal–CNS transmigration and subsequent progress of the infection.

## Results

### *S. suis* serotypes regulated CPS synthesis in different culture phases

A clinical isolate of *S. suis* serotype 2 wild-type (05ZYH33) was cultured and collected in the lag, logarithmic (log), and stationary phases (optical density at 600 nm [$OD_{600}$] of 0.20, 0.60, and 0.80, respectively) (*Figure 1A*). The CPS of 05ZYH33 collected in different growth phases was quantitated based on hydrophobicity in a bacterial adhesion to hydrocarbon (BATH) assay. High surface hydrophobicity leads to high bacterial adherence to hydrocarbon (high BATH value) and reflects low levels of CPS on the surface of 05ZYH33. As shown in *Figure 1B*, the CPS-deficient strain Δ2BSS2, used as a control, exhibited the highest BATH value (69.5 ± 6.51%). A similarly high BATH value (62.5 ± 9.73%) was found in the log phase of 05ZYH33. However, the BATH values in the lag and stationary phases of 05ZYH33 were substantially lower (37.0 ± 17% and 15.6 ± 9.2%, respectively). These data indicate that CPS of 05ZYH33 was synthesized dynamically in vitro, with the highest quantity in the stationary phase and the lowest in the log phase.

To determine the dynamic CPS synthesis in *S. suis* serotypes, four other serotypes of *S. suis* were cultured and collected in the log and stationary phases for CPS quantitation by BATH assays. As shown in *Figure 1C*, all the serotype strains had high hydrophobicity at the log phase and low at the stationary phase. The similarity in CPS patterns of these strains suggests a shared regulation mechanism by *S. suis* serotypes.

### 05ZYH33 in different cultural stages exhibited altered abilities for host cell adherence and resistance to bacterial killing

05ZYH33 and Δ2BSS2 were cultured and collected at the log and stationary phases to determine the effects of CPS on the binding of 05ZYH33 to epithelial cells. The adherence assay showed that while 16.5 ± 5.2% of Δ2BSS2 was associated with epithelial cells (HEp-2), only 4.96 ± 3.26% of 05ZYH33 in the log phase and 2.05 ± 1.58% of 05ZYH33 in the stationary phase were associated with the cells (*Figure 2A*). The bacterial binding was Δ2BSS2>05ZYH33 in the log phase, >05ZYH33 in the stationary phase. Similar binding patterns of 05ZYH33 and Δ2BSS2 were also observed in the assay using human brain microvascular endothelial cells (hBMECs), the major component of the BBB (*Figure 2B*). These results support that CPS reduces the adherence of the bacteria to host cells.

CPS of *S. suis* plays a critical role in the resistance to bacterial killing. 05ZYH33 in different growth phases was co-incubated with the blood from naïve mice to determine the effect of CPS on bacterial killing. After incubation for 3 hr, surviving bacteria in the blood examined by colony-forming units (CFUs) showed that substantially more 05ZYH33 in the log and stationary phases were detected compared with Δ2BSS2 in the matching growth phases. In addition, there were more surviving 05ZYH33 in the stationary phase compared with that in the log phase (*Figure 2C*). These results suggest a correlation between lower CPS in the NALT and a greater capacity for cellular association, whereas elevated CPS levels in the blood are linked to improved resistance against bactericidal activity. However, the mechanisms behind these associations remain unknown.

### The course of 05ZYH33 infection in mice following intranasal inoculation

The dynamic synthesis of CPS associated with differential adherence to host cells and resistance to bacterial killing in vitro suggested that 05ZYH33 regulates CPS synthesis for pathogenesis and survival

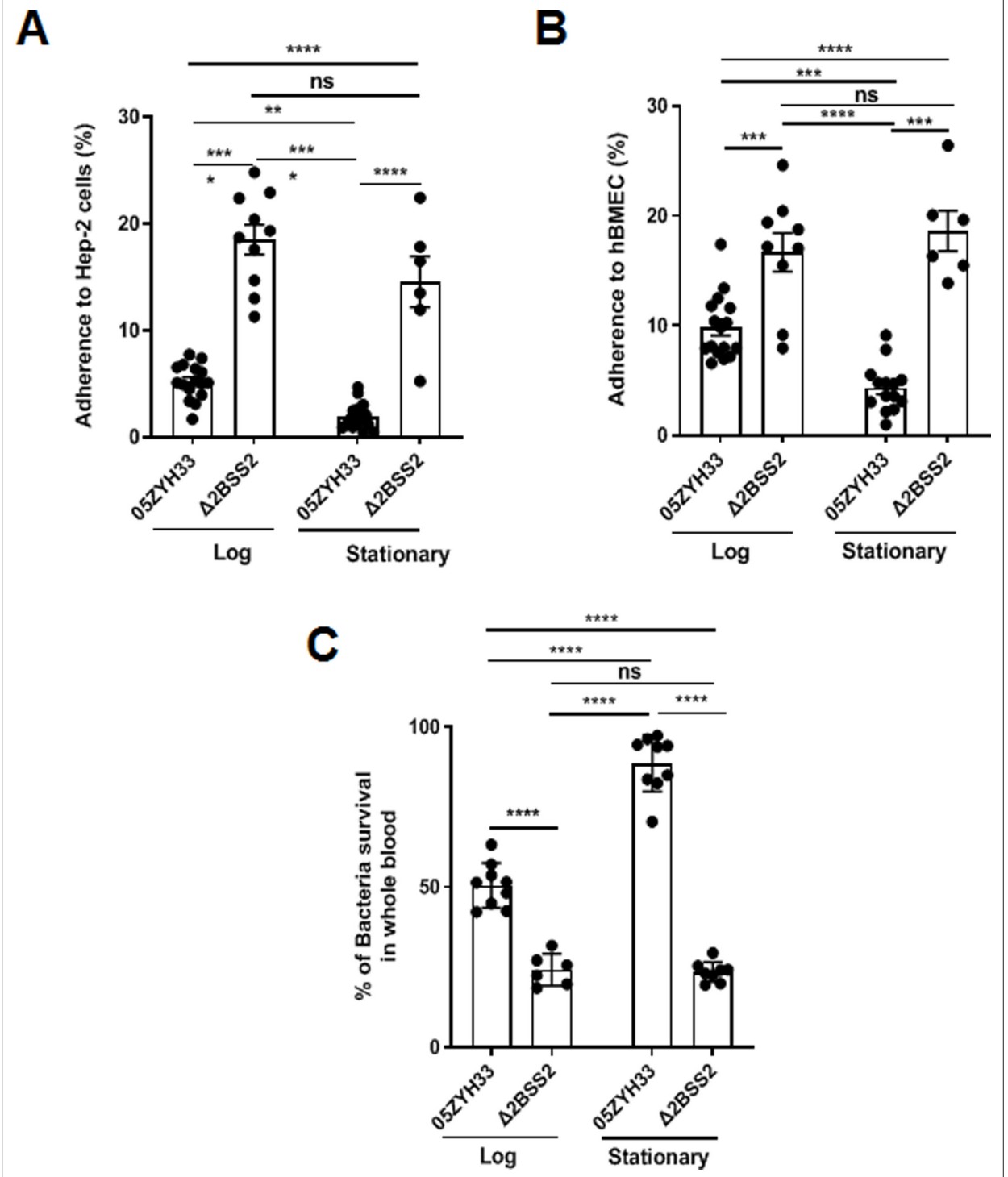

**Figure 2.** The adherence of 05ZYH33 to epithelial cells and its resistance to bacterial killing. HEp-2 or human brain microvascular endothelial cells (HBMECs) cells were inoculated with 05ZYH33 or Δ2BSS2 in either the log or stationary phase at an infection MOI of 10. (**A**) Adherence rate of *S. suis* to HEp-2 cells. (**B**) Adherence rate of *S. suis* to HBMECs. (**C**) *S. suis* strains in log and stationary phases were incubated with whole blood from naive mice for a duration of 3 hr, and the viability of *S. suis* was determined. Data are from three independent experiments and presented as means ± SEM (n=6–16). **p<0.01, ***p<0.001, ****p<0.0001; ns, not significant.

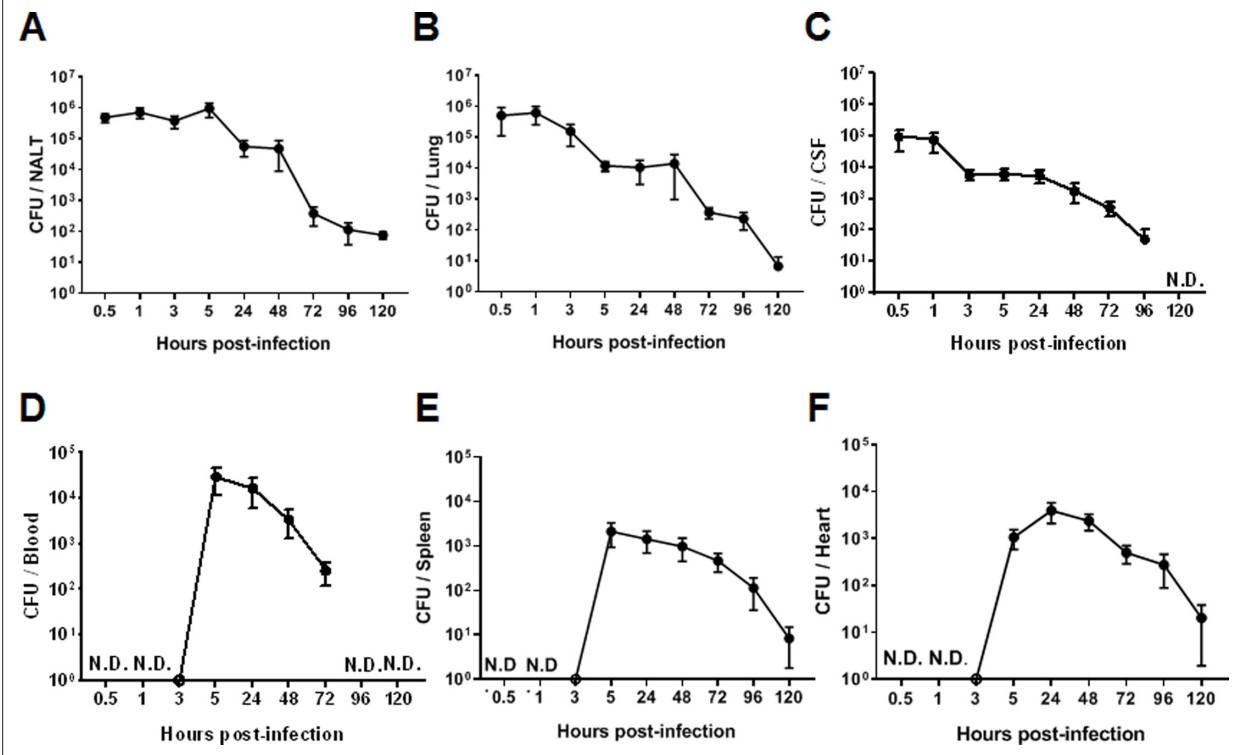

**Figure 3.** The intranasal infection course in mice. Mice were intranasally inoculated with 05ZYH33. Total colony-forming units (CFUs) in various bodily compartments were assessed at the indicated time points after inoculation, (**A**) nasal-associated lymphoid tissue (NALT), (**B**) the cerebrospinal fluid (CSF), (**C**) the lung, (**D**) the blood, (**E**) the spleen, and (**F**) the heart. The data presented are derived from two independent experiments and presented as means ± SEM (n=6). ND, not detected.

in a host. To determine the role of CPS dynamic synthesis in 05ZYH33 pathogenesis, mice were i.n. inoculated with 05ZYH33 to simulate *S. suis* natural infection in pigs, and CFUs in various bodily compartments were assessed at different points over 120 hr. By 0.5 hr after inoculation, CFUs of 05ZYH33 in the NALT, a pig tonsil homologue and the major niche for *S. suis* in pigs, and the lungs were similar ($5 \times 10^5$) and gradually reduced with time (*Figure 3A and B*), suggesting that the CFUs in the lungs are from the NC. Up to $10^5$ CFUs were also found in CSF samples by 0.5 hr (*Figure 3C*). The curves of CFUs in NALT, the lungs, and CSF showed a similar trend, which was a high level for a few hours followed by a gradual reduction with increasing days. In contrast, 05ZYH33 was not detected in the blood until 5.0 hr after inoculation (*Figure 3D*). As expected, CFUs in the spleen and heart at indicated times (*Figure 3E and F*) were similar to those in the blood, indicating that they disseminated from the circulation. These findings suggested that 05ZYH33 in the NC can transmigrate to the lungs and CNS before bacteremia and disseminate to other organs after entering the blood. During the 5 days (120 hr) following inoculation, the mice did not show significant disease symptoms, and their physical conditions were similar compared to non-infected mice (body weight, appetite, and activity). No obvious disease symptoms might be due to the bacteria cleared relatively quickly.

## The CPS of 05ZYH33 was synthesized differentially in stages of infection in mice

05ZYH33 was isolated from the bodily compartments of mice 12 hr after inoculation. The 12 hr of infection allow the detection of the bacteria in different body compartments of the infected mice. Following enrichment by culturing the isolates in TSB to an $OD_{600}$ of 0.7, CPS synthesis in vivo was determined by BATH assay. The hydrophobicity of isolates from NALT and the CSF was lower compared with that of Δ2BSS2, but higher than that of the isolates from the blood (*Figure 4A*). This indicates that 05ZYH33 downregulates CPS synthesis when it colonizes the nasopharyngeal mucosa or enters the CNS and upregulates CPS synthesis upon entering the blood. Transmission electron

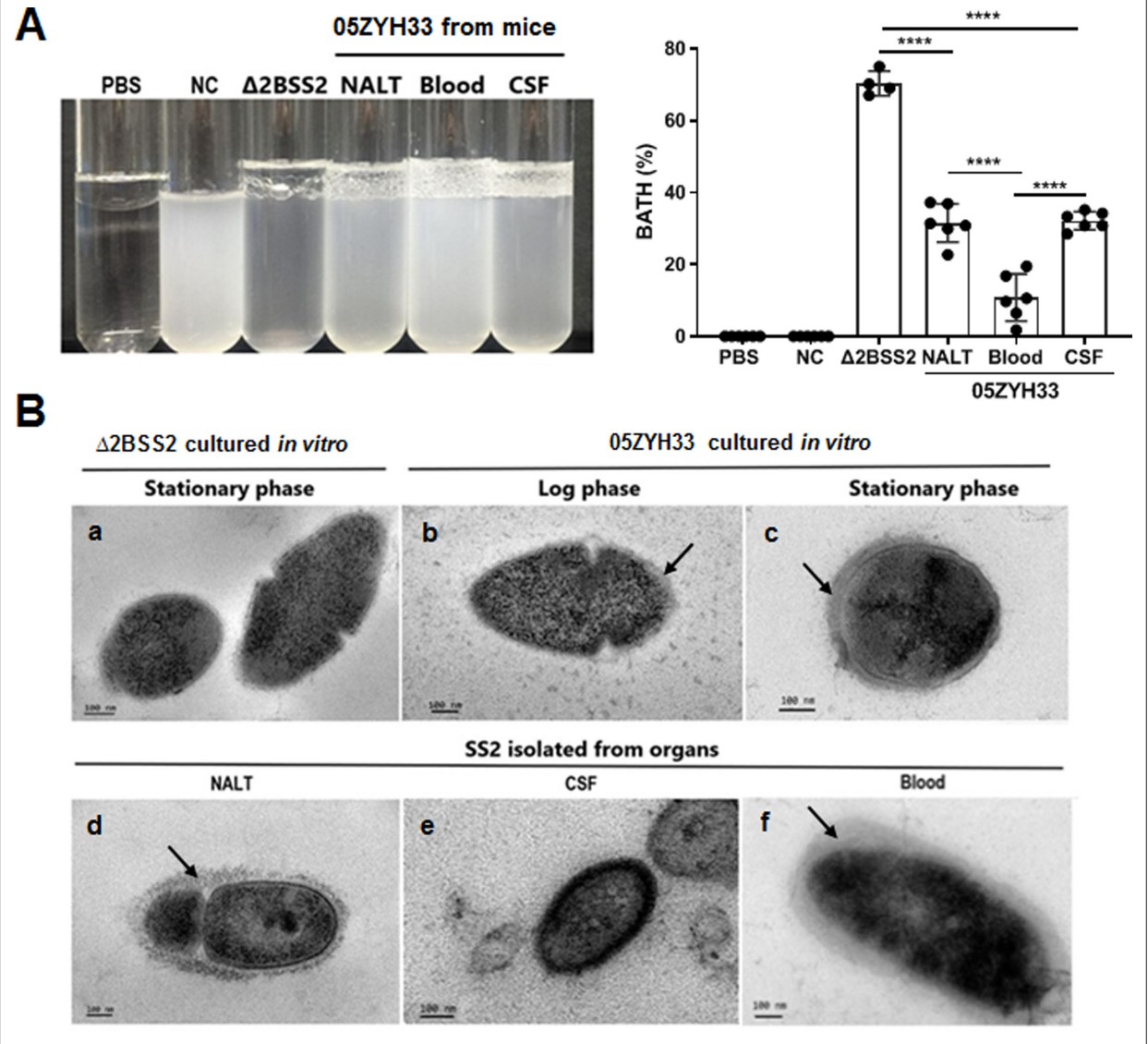

**Figure 4.** The capsular polysaccharide (CPS) of 05ZYH33 isolated from various bodily compartments was evaluated following infection. 05ZYH33 was intranasally inoculated in mice. Twelve hours after inoculation, colony-forming units (CFUs) in nasal-associated lymphoid tissue (NALT), the cerebrospinal fluid (CSF), and the blood were assessed for (**A**) hydrophobicity of bacterial cells by bacterial adhesion to hydrocarbon (BATH) assay and (**B**) CPS by transmission electron microscopy (TEM). Arrows indicate the CPS layers. Scale bar, 100 nm. Data in (**A**) are from two independent experiments and presented as means ± SEM (n=4–6). Nasal cavity (NC), 05ZYH33 suspension without hydrocarbon. ****p<0.0001.

microscopy (TEM) was performed directly on 05ZYH33 isolated from mice without in vitro culture to confirm the BATH assay results. For cultured strains without mouse infection (*Figure 4Ba, b, and c*), similar to Δ2BSS2 (*Figure 4Ba*), in the log phase (*Figure 4Bb*), there was no CPS observed on the dividing 05ZYH33 cell (left and larger cell), and only a thin and partial layer of CPS on the smaller dividing cell (right, smaller cell) was observed. Whereas, a thicker layer of CPS was observed on the surface of the bacteria in the stationary phase (*Figure 4Bc*). For 05ZYH33 isolated directly from infected mice (*Figure 4Bd, e, and f*), the CPS was much thicker, denser, and uniformly distributed in the blood isolates (*Figure 4Bf*) compared with the NALT isolates with thinner and much sparser CPS layer on the surface (*Figure 4Bd*). In contrast, the outermost layer of the isolates from the CSF was highly dense without an apparent CPS layer (*Figure 4Be*). These findings support that CPS downregulation is associated with efficient colonization and upregulation with increased resistance to bacterial killing in the blood.

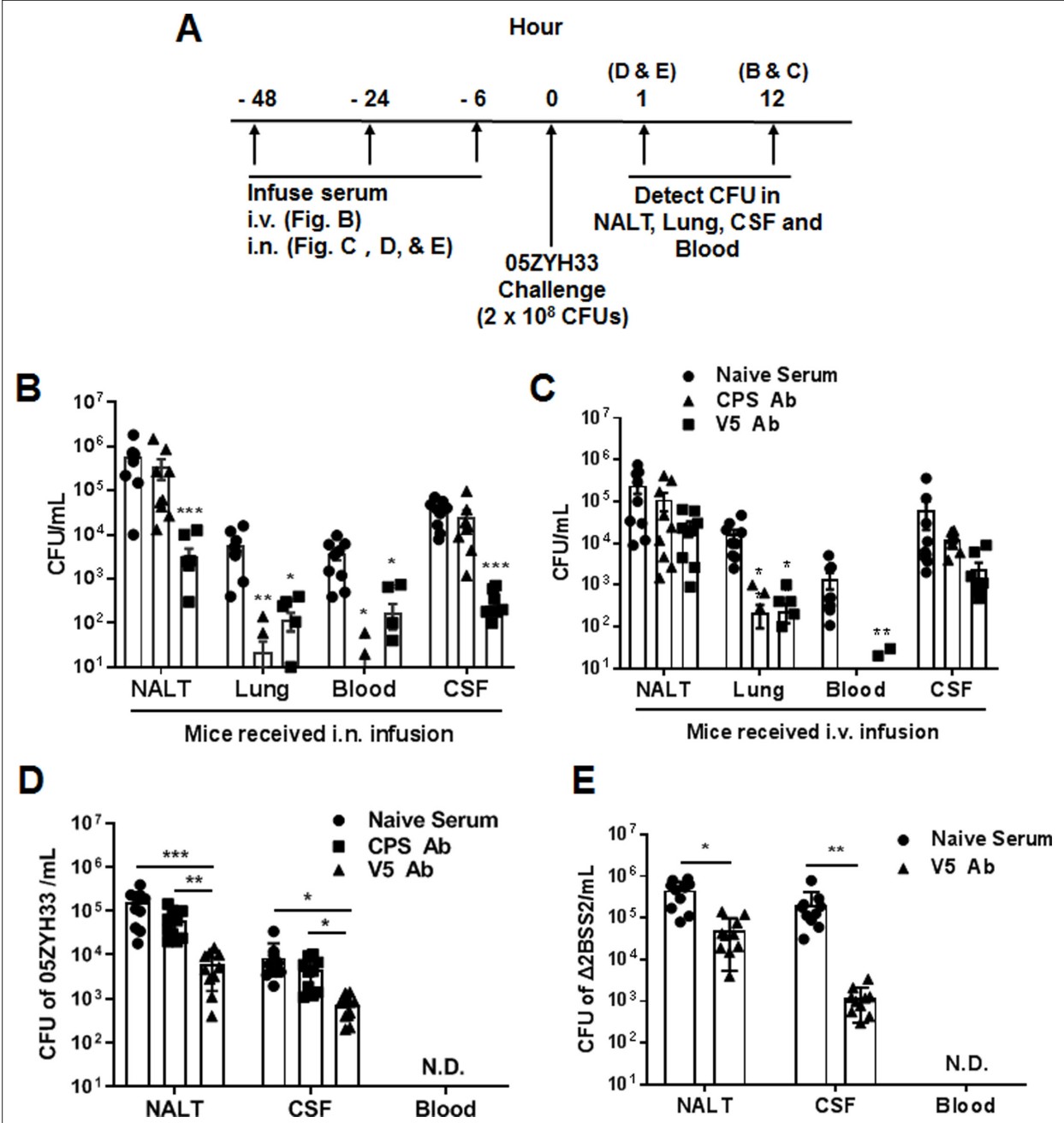

**Figure 5.** The role of anti-capsular polysaccharide (CPS) or anti-V5 serum in 05ZYH33 infection. (**A**) Schematic illustration of passive transfer of antisera and infection in mice. (**B, C**) Anti-V5 or anti-CPS antiserum was transferred through the intranasally (i.n.) or intravenous (i.v.) route. Six hours later, mice were i.n. challenged with 05ZYH33, then euthanized 12 hr post-challenge. Colony-forming units (CFUs) of 05ZYH33 in nasal-associated lymphoid tissue (NALT), the lungs, the blood, and the cerebrospinal fluid (CSF) were determined. (**D, E**) Mice were infused with anti-V5 or anti-CPS serum through the i.n. route and inoculated i.n. with 05ZYH33 or Δ2BSS2 6 hr later. One hour later, mice were euthanized, and CFUs in NALT, the blood, and the CSF were determined. Data are the mean ± SEM of three independent experiments (n=10). *p<0.05, **p<0.01, ***p<0.001.

## An antiserum against non-CPS virulence factors of *S. suis* reduced 05ZYH33 colonization in NALT, CSF, and lungs

We have previously demonstrated that immunization with multiple non-CPS virulence factors of *S. suis* (named V5) prevents 05ZYH33 colonization of the nasal mucosa and neuropathological symptoms in mice (*Xing et al., 2019*). We hypothesized that the serum against V5 would inhibit the nasal–CNS transmigration of 05ZYH33. Mice were i.n. infused with anti-V5 or anti-CPS serum, inoculated i.n.

with 05ZYH33 6 hr later, then euthanized 12 hr post-inoculation (*Figure 5A*) to detect the bacteria in different body compartments of the infected mice. In mice that received antisera through the i.n. route, CFUs reduced in all examined compartments of anti-V5 recipients but only reduced in the lungs and blood of anti-CPS recipients (*Figure 5B*). These results indicated that 05ZYH33 survival in the blood depends on CPS, and nasopharyngeal colonization and nasal–CNS transmigration are more dependent on non-CPS molecules of 05ZYH33. In addition, when mice received either anti-V5 or anti-CPS through the intravenous (i.v.) route, CFUs were reduced in the lungs and blood but not in NALT and the CSF (*Figure 5C*), compared with mice that received naïve serum through the i.v. route. These observations suggest that systemic immunity could efficiently protect against infection in the lungs and blood but not against colonization in the nasal mucosal and CNS, at least at the early infection time.

The simultaneous 05ZYH33 colonization in NALT and CNS supports a nasal–CSF transmigration. To exclude the possibility that CFUs in the CNS at 12 hr post-infection are disseminated from the circulation, mice received antisera via the i.n. route and were inoculated with 05ZYH33 as before; then, CFUs were determined 1 hr post-inoculation before the bacteria entered the blood (*Figure 5A*). As shown in *Figure 5D*, CFUs in NALT and the CSF of anti-CPS recipients were similar to those in naïve-serum recipients; however, those in NALT and the CSF of anti-V5 recipients were 10-fold lower compared with those of the naïve-serum recipients. To confirm that the nasal–CNS transmigration is independent of CPS, mice received naïve serum or anti-V5 via the i.n. route, then were inoculated with Δ2BSS2. One hour later, we found $4.5\times10^5$ CFUs in NALT and $2.1\times10^5$ CFUs in the CSF of naïve-serum recipients. However, CFUs were 10-fold and 100-fold lower in NALT and the CSF, respectively, of anti-V5 recipients compared with those in naïve-serum recipients (*Figure 5E*). These results demonstrate that 05ZYH33 in the CNS in the early infection is from the NC independent of CPS.

## The olfactory nerve is the invasion route of 05ZYH33 for nasal–CNS transmigration

The olfactory nerve is a port of entry to the meninges for some human meningeal pathogens and is likely the route of the early CNS dissemination by 05ZYH33. One hour after infection, the mice were sacrificed, and their heads were sectioned and stained for histological analysis. *Figure 6A* presents a histological overview of the anatomical structures at the distal NC and olfactory bulb (OB) regions, while *Figure 6B* illustrates the locations of the anatomical compartments in the area. The tissue section showed that compared with uninfected mice (*Figure 6Aa, c, and e*), scattered and grouped bacteria were observed on the surface of and in the olfactory epithelium (OE) (*Figure 6Ab and d*, orange signal, arrowheads), accompanied by severely destroyed tissue (indicated by the loss of immunofluorescence signal in the area) (*Figure 6A and d*). Large groups of bacteria were observed within the OB (*Figure 6A and f*). These observations suggest that 05ZYH33 in the NC could invade the OE through the sensory neurons protruding into the NC and subsequently enter the brain along the sensory neurons at the cribriform plate. Histological analysis of the brain in 05ZYH33-infected mice revealed that inflammatory cells were infiltrated in the distal part of the brain tissue with surrounding edema at 1 hr post-inoculation (*Figure 6Cc and d*). The pathological changes were more severe 9 days after 05ZYH33 inoculation (*Figure 6Ce and f*). Since CSF flows into the subarachnoid space of the spinal cord and around the brain (*Yoon et al., 2024*; *Spera et al., 2023*), the distal inflammation is consistent with the brain's anatomical structure. Thus, nasal–CNS transmigration could be an efficient route of *S. suis*-mediated CNS infection.

Mice were pretreated with acetic acid to increase the sensitivity to 05ZYH33 infection (*Xing et al., 2019*). To exclude that nasal–CNS transmigration results from irritation of the nasopharyngeal mucosa by acetic acid, mice were infected without acid treatment (pretreated with phosphate-buffered saline [PBS]). One hour post-inoculation, CFUs were found in the CSF of the mice even though the numbers were lower than those of acidic acid-treated mice (*Figure 6D*). CFUs were also detected in the CSF when mice were infected with Δ2BSS2 without acid treatment (*Figure 6E*). These results indicated that nasal–CSF transmission takes place without acidic acid and independent of CPS, and that irritation of the nasopharyngeal mucosa facilitates nasal–CNS transmission.

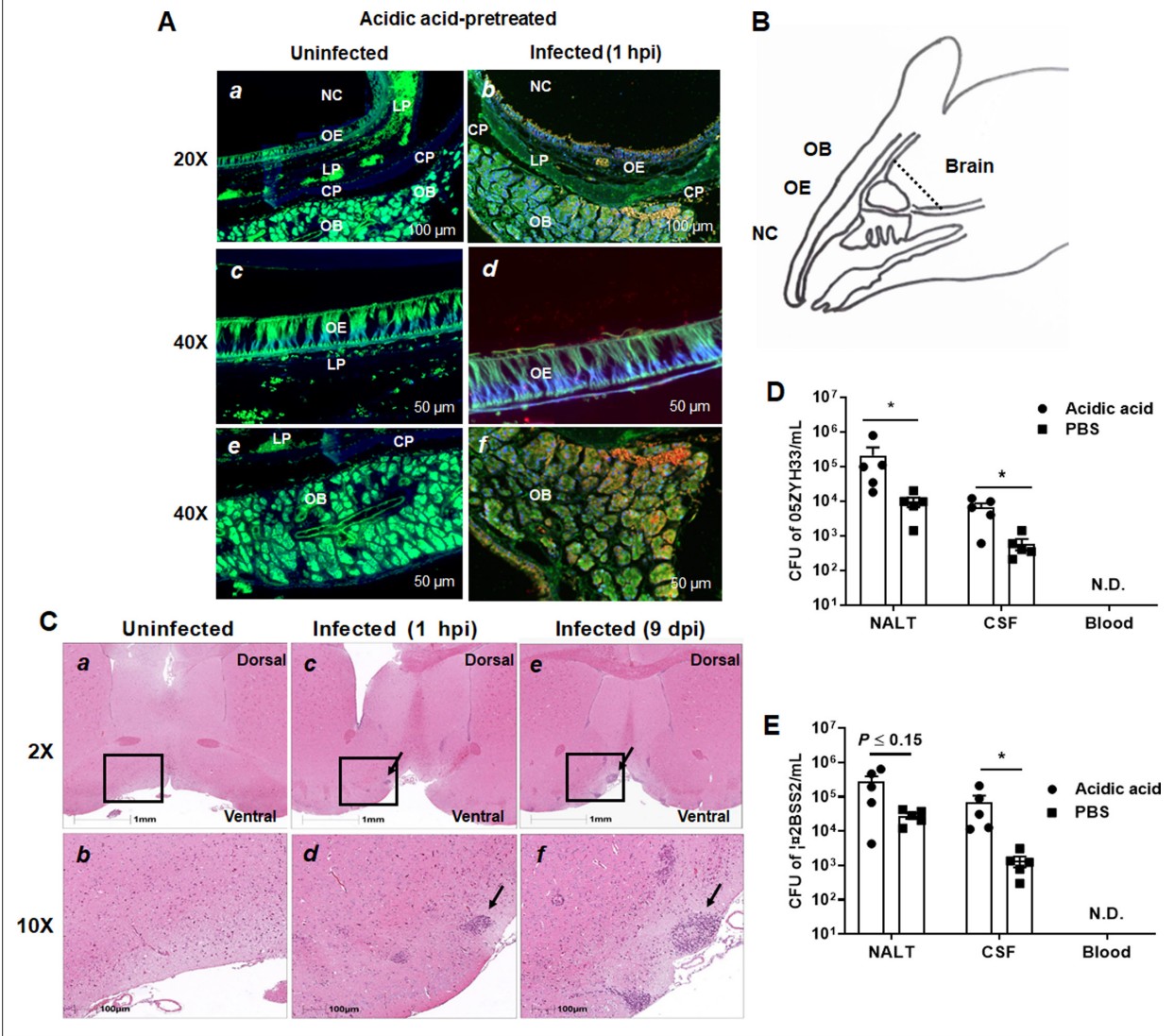

**Figure 6.** Detection of the presence of 05ZYH33 in the tissue of olfactory nerve area. Mice were inoculated with acetic acid through the nostril, and 1 hr later, infected intranasally (i.n.) with 05ZYH33. (**A**) Sagittal views of the olfactory system. Sections of the distal nasal cavity and olfactory epithelium from uninfected (a, c, and e) and 05ZYH33-infected (b, d, and f) mice. Red or orange, 05ZYH33; green, the neuronal marker β-tubulin III; blue, DNA. NC, nasal cavity; OE, olfactory epithelium; LP, lamina propria; CP, cribriform plate; OB, olfactory bulb. (**B**) A schematic drawing of the sagittal plane of the rodent nose elucidates the compartments of the olfactory bulb (OB), olfactory epithelium (OE), nasal cavity (NC), and brain. This panel is redrawn from 'Mouse Olfactory System' (inspiredpencil.com). The dotted line indicates the anteroposterior localization of the coronal sections in C. (**C**) One hour or nine days after 05ZYH33 infection, coronal brain sections were prepared and stained with hematoxylin and eosin (H&E). The sections display the regions of the ventral striatum and basal forebrain located behind the anterior olfactory nucleus. Arrows indicate infiltrated inflammatory cells in the lower areas of the ventral striatum or basal forebrain. (**D, E**) Mice were inoculated with acetic acid or phosphate-buffered saline (PBS), and 1 hr later infected i.n. with 05ZYH33 or Δ2BSS2. (**D**) Colony-forming units (CFUs) of 05ZYH33 or (**E**) Δ2BSS2 in nasal-associated lymphoid tissue (NALT), the cerebrospinal fluid (CSF), and blood were determined 1 hr after infection.

## Discussion

CPS is an important virulence factor of *S. suis*, but its role in *S. suis* pathogenesis is controversial. Most animal models in studies of *S. suis* employ the intraperitoneal or i.v. route for infection, which bypasses the nasopharynx, the natural infection route of *S. suis*. In this study, CPS thickness of *S. suis* SS2 strain 05ZYH33 was monitored in an i.n. infection mouse model. The results indicate that *S. suis* CPS can be dynamically synthesized during the development of the bacterial infection. We showed that the downregulation of CPS synthesis by 05ZYH33 is associated with increased nasopharyngeal colonization and early nasal–CNS transmigration before bacteremia. This is consistent with previous

studies that showed the upregulation of CPS is related to increased resistance to bacterial killing later in the blood. The study also indicates that CPS-independent immunity can prevent nasal–CNS transmigration and the progression of 05ZYH33 infection.

Clinical CNS symptoms in pigs are often preceded by a phase of bacteremia (*Arends and Zanen, 1988*; *Meng et al., 2015*), indicating that *S. suis* meningitis results from the dissemination of bacteria from the blood through the BBB or the blood–CSF barrier. Our study revealed that, like some human meningeal pathogens, the olfactory nerve epithelium provides an alternative route for 05ZYH33 invasion to the CNS. Swine influenza virus (SIV) infection was reported to promote *S. suis* invasion of deeper tissues by damaging epithelial cells (*Meng et al., 2015*; *Meng et al., 2019*; *Wu et al., 2015*). Nasopharyngeal inflammation by co-infected *S. suis* and SIV could cause tissue damage to the OE, which may exacerbate nasal–CNS entry. As the mouse is not a natural host of *S. suis* and is insensitive to the infection, a relatively high dosage is needed for optimal bacterial retention in NALT, which can lead bacteria into the lungs. Thus, early 05ZYH33 colonization in the lungs of this model may not reflect the state of *S. suis* infections in pigs. The early 05ZYH33 colonization in NALT and the CSF with similar dynamics suggests that meningitis could be early manifestations of *S. suis* disease before multi-organ dissemination through the blood. The findings in the study indicate that the restriction of *S. suis* proliferation in NALT is critical to prevent the severity and lethality of *S. suis* disease.

We noticed that the outermost layer of the isolates from the CSF was highly dense without an apparent CPS layer (*Figure 4Be*), suggesting that other membrane components of the CSF isolates are also changed besides CPS. It is unclear whether this change is related to CSF colonization and what molecules have changed. Although CPS plays an important role in *S. suis* survival in the blood, it is not the only component required for the event (*Berthelot-Hérault et al., 2001*; *Berthelot-Hérault et al., 2005*). Similar to that observed with anti-CPS, infusion of anti-V5 through the i.v. route reduced 05ZYH33 in the blood and lungs, indicating that blocking CPS-independent mechanisms can restrain the progression of 05ZYH33 infection and suggests that non-CPS virulence factors, such as those in V5, are required for 05ZYH33 survival in the blood and lungs. Furthermore, although anti-CPS promoted 05ZYH33 clearance in the blood, it did not prevent 05ZYH33 colonization of the NC and invasion of the CNS. However, anti-V5 reduced 05ZYH33 in all examined compartments when administered via the i.n. route, supporting that inhibition of nasal colonization by CPS-independent immunity is critical to prevent the progression of the disease. Moreover, the nasal–CNS route may not be the main pathway for clinical meningitis compared to the blood–CNS route. However, mucosal blockage of nasal colonization reduces CFU in the CSF more effectively than systemic antibody, indicating that reduction of nasal colonization at early time would significantly prevent the bacteria from entering the CNS through the early nasal and later blood route.

In summary, this study supports an updated *S. suis* infection model. Briefly, 05ZYH33 downregulates CPS to colonize the URT mucosa, where it can transmigrate to the CNS through the OE early in the infection process. 05ZYH33 upregulates CPS to avoid bacterial killing when it enters the blood, leading to multi-organ dissemination. The regulation of CPS at different stages of 05ZYH33 pathogenesis and the protection offered by CPS-independent immunity provide valuable information for clinical treatment and development of vaccines across serotypes of *S. suis*.

# Materials and methods
## Bacterial strains and growth conditions

*S. suis* strain SS2 05ZYH33 (GenBank no. for complete genome sequence: CP000407.1) was isolated from a fatal human case of streptococcal toxic shock-like syndrome in 2005 in Sichuan province, China. A capsular mutant of *S. suis* 05ZYH33 (Δ2BSS2) was obtained from Prof. Jia-Qi Tang (General Hospital of the Eastern Theater of the People's Liberation Army, China) (*Cao et al., 2011*). *S. suis* serotypes SS3, SS7, SS9, and SS1/2 were gifts from Dr. Shengbo Cao (National Key Laboratory of Agricultural Microbiology, Huazhong Agricultural University, Wuhan, China). The information on these strains is listed in *Supplementary file 1*. All *S. suis* strains were cultured in tryptic soy broth (TSB) containing 5% newborn bovine serum (FBS) or on tryptic soy agar supplemented with 5% FBS at 37°C and 5% $CO_2$. The cultured strains were washed and resuspended in PBS. CFUs were verified by plating on blood agar.

## Growth curve of 05ZYH33

05ZYH33 was grown in TSB-FBS medium to an $OD_{600}$ of approximately 0.4 and then diluted 1000 times with fresh medium. Next, 200 µL of the bacterial suspension was added in quadruplicate to a 100-well honeycomb plate and cultured at 37°C and 5% $CO_2$ for 16 hr. The $OD_{600}$ of each well was measured at different times using a CMax Plus plate reader (Molecular Devices, San Jose, CA, USA).

## Determination of CFUs in various bodily compartments

Mice were infected i.n. with 05ZYH33 (stationary phase). Then, at different time points, the lung, spleen, heart, and NALT were taken from mice, and the blood and CSF were collected. A single-cell suspension of each organ was prepared in 1–5 mL PBS. CFUs in the single-cell suspension, the blood, and the CSF were determined by plating serially diluted samples on blood agar plates and expressed as total CFUs in each organ or in the blood and CSF by the following formula: CFUs on the plate × dilution factor × total volume of a sample.

## BATH assay

The hydrophobicity of *S. suis* was assessed by a modified BATH assay (*Shuster et al., 2019*). Briefly, strains 05ZYH33, Δ2BSS2, and other serotypes of *S. suis* cells in cultures were collected, washed, and resuspended in PBS, adjusting the $OD_{600}$ of the suspensions to 2.0 (±0.2) with PBS. Four milliliters of each suspension were added to a tube containing 300 µL xylene and 300 µL hexadecane. A negative control comprised of a 4 mL bacterial suspension without hydrocarbon. After vortexing for 20 s, the tubes were maintained at 37°C for 30 min. The $OD_{600}$ of the resulting lower aqueous layer and the tube contents before vortexing were measured. The ratio of the absorbance of the bacteria assay tubes (Ab) to the absorbance of the bacteria control (Ac) was calculated as adhesion to hydrocarbon (%) = (Ac − Ab)/Ac◊100.

## Cell lines

The human laryngeal epithelial cells (HEp-2, CL-0104) were purchased from Procell Life Science & Technology Co., Ltd. Wuhan, Hubei Province, and cultured according to the manufacturer's instruction. The hBMECs were kindly provided by Prof. Xiang-Ru Wang in Huazhong Agricultural University, Wuhan, Hubei, China, and cultured as described by *Yang et al., 2016*.

## Bacterial adherence assay

The human laryngeal epithelial cells (HEp-2, CL-0104) were purchased from Procell Life Science & Technology Co., Ltd. Wuhan, Hubei Province, and cultured according to the manufacturer's instruction. The hBMECs were kindly provided by Prof. Xiang-Ru Wang in Huazhong Agricultural University, Wuhan, Hubei, China, and cultured as described by *Yang et al., 2016*. Adherence of 05ZYH33 or Δ2BSS2 in log and stationary phases was performed as described previously with slight modifications (*Li et al., 2015*). Cells were cultured in 24-well plates to 80–90% confluence (~$1.2 \times 10^5$ cells/well), then starved overnight in DMEM without serum and antibiotics. Cell monolayers were washed with PBS and overlaid with DMEM with 10% FBS (500 µL/well) containing 05ZYH33 or Δ2BSS2 (MOI = 10). After incubation for 2 hr at 37°C and 5% $CO_2$, unbound bacteria were removed, and the cells were lysed with 0.025% Triton X-100. CFUs in samples of diluted cell lysate were determined on blood agar plates and expressed as the percentage of adherence: (CFUs in wells with cells/CFUs in wells without cells)×100.

## Bactericidal assay

Bactericidal assays were performed as described by *Pian et al., 2012*, with slight modifications. Briefly, $1 \times 10^5$ 05ZYH33 or Δ2BSS2 in log and stationary phases were added to whole blood from naïve mice. The mixtures were rotated for 3 hr at 37°C and 5% $CO_2$. A set of the same bacterial samples was added to naive serum to determine bacterial numbers in the 3 hr incubation without killing. Viable bacterial counts were determined by plating diluted samples on blood agar. The anti-killing capacity was calculated as (CFU in blood/CFU in serum)×100%.

## Mouse infection

C57BL/6JCnc wild-type mice (aged 6–7 weeks) were purchased from Vital River Laboratory Animal Center (Beijing, China) and randomly assigned to groups. Mice were pretreated with 5 µL of 1% acetic

acid (pH 4.0) or PBS per nostril 1 hr before intranasal infection (*Xing et al., 2019*). After further anesthesia, mice were inoculated i.n. with 05ZYH33 (stationary phase, $2 \times 10^8$ CFU/10 µL). Five µL of the inoculum was delivered i.n. to each nostril (total 10 µL). This volume allows optional bacterial retention in NALT.

## Specimen collection and preparation

Mice were euthanized, then blood samples were collected from the heart, and the lungs, heart, spleen, and NALT were removed and washed with PBS to eradicate blood contamination. Single-cell suspensions of the tissues were prepared by mechanical dissociation of tissue through a 100 µm nylon mesh in 1 mL PBS. For CSF collection, the nose part was removed from the head along the line of the eyeballs after the collection of NALT tissue. The brain tissue was taken from the remaining part of the head and soaked in 1 mL PBS for 1 hr after sufficient laceration to expose the subarachnoid space without mincing. This operation avoided damaging the integrity of brain tissue as much as possible. Supernatants of the single-cell suspension, and the blood and CSF samples were used to detect CFU and examine the CPS.

## Transmission electron microscopy

*S. suis* cells were enriched by gradient centrifugation of homogenized NALT and samples of the blood and CSF of infected mice. After washing, the bacteria were fixed in 2% paraformaldehyde supplemented with 0.05% ruthenium red in 0.1 M phosphate buffer (pH 7.0) for 30 min at room temperature. The bacterial samples were further prepared and embedded in epoxy resin (Epon 812, Merck) as described by *Bojarska et al., 2020*. Ultrathin (70–80 nm) sections were cut, stained with 1.2% uranyl acetate and 2.5% lead citrate, and then examined with a JEM-1400 (120 kV) transmission electron microscope (JEOL, Tokyo, Japan). Images were taken with a Morada digital camera (Olympus SIS, Münster, Germany) at 10 M pix resolution.

## Cloning and expression of recombinant V5 proteins

V5 contains five conserved virulence factors of *S. suis*. They are SrtA (amino acids 50–249), ScpCL (amino acids 36–535), MRP (amino acids 283–721), SLY (amino acids 28–497), and SspA (amino acids 41–852), and prepared as described by *Xing et al., 2019*. Briefly, the genes of V5 were cloned from the genome of *S. suis* strain A7 (serotype 2) (GenBank accession number: NC_017622.1). The recombinant proteins of V5 were expressed in *E. coli* BL21 (DE3) and purified by immobilized metal ion affinity chromatography to reach 95% purity. Bacterial lipopolysaccharide was removed following purification (≤0.1 EU/µg). Protein concentrations were determined using the BCA Protein Assay Kit according to the manufacturer's instructions (TIANGEN, catalog no. PA115-02).

## Preparation and purification of anti-CPS and anti-V5 antibodies

A female New Zealand white rabbit (SPF, 2.0–2.5 kg) was purchased from JinMuYang Laboratory Animal Center (Beijing, China) and kept in the SPF animal facility for 1 week. V5 (1 mg) was mixed with 1 mL incomplete Freund's adjuvant (Becton Dickinson, Franklin Lakes, NJ, USA) and subcutaneously injected into the rabbit four times at 1 week intervals. Blood samples were harvested 10 days after the last immunization, and the serum was isolated. Anti-CPS (anti-*S. suis* serotype 2) serum was purchased from SSI Diagnostica (Hillerød, Denmark) and purified by absorption with V5 by Ni-NTA agarose (QIAGEN, catalog no. 30210) according to the manufacturer's instructions. Briefly, 200 µL of V5 (His-tagged, 1 µg/µL) were loaded on the 5 mL Ni-NTA agarose column. The collected antiserum was further purified by incubation with formalin-inactivated Δ2BSS2 ($5 \times 10^{10}$ CFU) overnight at 4°C. The purified anti-CPS recognized 05ZYH33 but not Δ2BSS2 as determined by enzyme-linked immunosorbent assay (ELISA) against 05ZYH33 and Δ2BSS2, respectively. Titers of the purified antisera against V5 and CPS were evaluated by ELISA and stored at −80°C until use.

## Passive antiserum transfer

Female C57BL/6JCnc mice (6–8 weeks) were transferred with anti-CPS or anti-V5 serum through the i.n. or i.v. route. Optimal amounts of antisera were predetermined in mouse infection assays (the minimum amount of anti-V5 to inhibit 05ZYH33 colonization in NALT and the minimum amount of anti-CPS to prevent bacteremia). For anti-CPS serum, 20 µg/200 µL (100 µg/mL) was used for tail

vein injection or 5 µg/30 µL (170 µg/mL) for nasal drip. For anti-V5 serum, 100 µg/200 µL (500 µg/mL) and 50 µg/30 µL (170 µg/mL) were used through the i.v. and i.n. routes, respectively. Six hours after antiserum transfer, mice were inoculated via the i.n. route with $2 \times 10^8$ CFU of 05ZYH33, and CFUs in different compartments of the infected mice were determined.

## Immunofluorescence staining and confocal microscopy

Immunofluorescence staining and confocal microscopy of the NC were performed as described by *Pägelow et al., 2018*. Briefly, whole heads of the mice were fixed with 4% PFA for at least 24 hr before embedding in paraffin wax. Sections of longitudinally bisected heads were deparaffinized followed by antigen retrieval. Immunostaining was conducted using a primary rabbit anti-*S. suis* antibody (1:100) and a mouse anti-tubulin beta III antibody (Merck Millipore, Darmstadt, Germany catalog no. mab5564, 1:300) to detect neuronal structures, and then appropriate fluorophore-conjugated secondary antibodies: donkey anti-rabbit Cy3 (Jackson ImmunoResearch Laboratories, Ely, UK, catalog no. 711-166-152, 1:1000), donkey anti-mouse FITC (Jackson ImmunoResearch Laboratories, Ely, UK, catalog no. 715-096-150, 1:500). Tissues were mounted with Vectashield Mounting Medium with DAPI (Vector Laboratories, Burlingame, CA, USA, catalog no. H-1200). Images were captured using a Leica Aperio Versa 200 scanning microscope and color-balanced uniformly across the field of view using the Image Processing Leica Confocal and ImageJ Software (Wayne Rasband, National Institutes of Health, Bethesda, MD, USA).

## Histological analysis of brain tissue

Mice were treated with 5 µL of 1% acetic acid (pH 4.0) per nostril. One hour later, one group of mice was inoculated i.n. with 05ZYH33 ($2 \times 10^8$ CFU). A control group of mice was given PBS via the i.n. route. Infected mice were euthanized 1 hr and 9 days after infection, respectively. The brain was removed and quickly fixed with 4% paraformaldehyde. After paraffin embedding, tissue sections were stained with hematoxylin and eosin according to a standard protocol and examined using a Leica SCN400 slide scanner (Leica Microsystems, Germany).

## Statistical analysis

Statistical analyses were performed in GraphPad Prism software, version 7.0. CFUs were analyzed by two-tailed unpaired Mann–Whitney U nonparametric t-tests. An unpaired, two-tailed Student's t-test was used to analyze the statistical significance between two groups. One-way analysis of variance (ANOVA) with Tukey's post hoc test was used to analyze the statistical significance of differences among more than two groups. A p-value of <0.05 was considered significant.

## Acknowledgements

The authors acknowledge Prof. Jiaqi Tang of General Hospital of the Eastern Theater of the People's Liberation Army of China for providing the capsular mutant of *S. suis* 05ZYH33 (Δ2BSS2), Dr. Shengbo Cao (National Key Laboratory of Agricultural Microbiology, Huazhong Agricultural University, Wuhan, China) for sharing *S. suis* serotype strains, Prof. Xiang-Ru Wang in Huazhong Agricultural University, Wuhan, Hubei, China, for the hBMECs, and Dr. Junfeng Hao in the Pathological Laboratory of Institute of Biophysics, CAS, for performing the immunofluorescence microscopy. This research was supported by the National Natural Science Foundation of China grant 31872743 (to BW) and the China Postdoctoral Science Foundation Grant 2019M650875 (to XW).

## Additional information

### Funding

| Funder | Grant reference number | Author |
| --- | --- | --- |
| China Postdoctoral Science Foundation | 2019M650875 | Xingye Wang |

| Funder | Grant reference number | Author |
| --- | --- | --- |
| National Natural Science Foundation of China | 31872743 | Beinan Wang |

The funders had no role in study design, data collection and interpretation, or the decision to submit the work for publication.

## Author contributions

Xingye Wang, Conceptualization, Data curation, Supervision, Funding acquisition, Investigation, Methodology, Writing – original draft, Project administration, Writing – review and editing; Jie Wang, Data curation, Methodology; Ning Li, Xin Fan, Methodology; Beinan Wang, Formal analysis, Funding acquisition, Methodology, Writing – original draft, Writing – review and editing

## Author ORCIDs

Xingye Wang ⓘ https://orcid.org/0009-0001-0909-4838
Beinan Wang ⓘ https://orcid.org/0000-0002-9260-7197

## Ethics

This study was performed in strict accordance with the recommendations in the Guide for the Care and Use of Laboratory Animals of the IMCAS (Institute of Microbiology, Chinese Academy of Sciences) Ethics Committee. The protocol was approved by the Committee on the Ethics of Animal Experiments of IMCAS (permit no. APIMCAS2019004). Mice were bred under specific pathogen-free (SPF) conditions in the laboratory animal facility at IMCAS. All animal experiments were conducted under isoflurane anesthesia, and all efforts were made to minimize suffering.

Reviewer #1 (Public review): https://doi.org/10.7554/eLife.101760.3.sa1
Author response https://doi.org/10.7554/eLife.101760.3.sa2

# Additional files

## Supplementary files

Supplementary file 1. Information of the strains used in the study.
MDAR checklist

## Data availability

All data generated or analysed during this study are included in the manuscript can be found on Dryad.

The following dataset was generated:

| Author(s) | Year | Dataset title | Dataset URL | Database and Identifier |
| --- | --- | --- | --- | --- |
| Wang X, Wang J, Wang B | 2025 | Regulative synthesis of capsular polysaccharides in the pathogenesis of *Streptococcus suis* | https://doi.org/10.5061/dryad.0p2ngf2c1 | Dryad Digital Repository, 10.5061/dryad.0p2ngf2c1 |

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
