## [Editor Report · eLife Assessment]

This **useful** study uses a model of Streptococcus suis (a pig pathogen) infection in mice using an intranasal route, the natural route of infection ignored in most of the literature. The study aims to understand how capsular polysaccharides (CPS) contribute to neuropathology and virulence. The findings suggest that the olfactory route may lead to meningitis before bacteremia occurs and that CPS down-regulation may play a role in this process. However, the study remains **incomplete** as presented.

---

## [Referee Report · Reviewer #1 (Public review)]

Summary:

The manuscript by Wang et al. investigates the relationship between Streptococcus Suis (S. Suis) growth phases and levels of virulence factor, capsular polysaccharide (CPS), in the bacterial cell wall. They use an understudied mouse intranasal infection model to connect growth phase related CPS abundance to the pathogenicity of the bacteria in the nose, blood, and other organs. Adoptive transfer of serum against either CPS or V5 (five other virulence factors) reinforces their discovery of CPS levels on S. Suis in different organs and stages of infection. Vaccination against bacterial infections can be difficult, and understanding how the serotype of a bacterial pathogen changes between infection sights and systemic disease is critical. Further, understanding host-pathogen interactions at early time points in the upper respiratory tract may have broad implications for vaccine development. While some of the results are interesting and compelling, others are not supported by the data and require further experimental work.

Strengths:

The model of intranasal infection is compelling to expand upon work previously done in vitro and with systemic routes of infection. The histology and fluorescent imaging of the olfactory epithelium and olfactory bulb complement work in Figure 2 about the attachment of S. suis to epithelial cells and the bacterial burden over time in different organs of Figure 3. Histology was performed at 1 hour and 9 days after intranasal infection with stationary phase S. suis and drives home that this pathogen can invade the olfactory nerve and may potentially cause bacterial meningitis seen in some infected swine.

The adoptive transfer of either anti-CPS or anti-V5 to mice before infection at both longer (12 hr), and shorter (0.5 hr) time points is useful to demonstrate that the changes in cell wall composition between the NALT/CSF and blood compartments result in different efficacy in clearing bacteria from those locations. This is fundamental for the development of vaccines for the swine industry and begs those developing other bacterial vaccines to consider what virulence factors are the most useful as neutralizing antibody targets at the sight of bacterial invasion.

Demonstrating that the amount of CPS within the cell wall of S. Suis is related to the growth phase of the bacteria is an important consideration for vaccine development. While others had previously shown that CPS levels were higher in the blood than in the CNS, and that CPS decreases the invasion of epithelial cells, the close look at the olfactory epithelium at an early time point ties together in vitro findings. The control of a CPS-negative strain was critical to understanding their findings. The location and the microbial community that bacterial pathogens live within may change the growth phase and therefore also the cell wall components.

Weaknesses:

The authors present compelling data that is relevant to the development of anti-bacterial vaccinations and show a relationship between CPS levels and pathogenicity. However, the use of a laboratory murine model requiring acetic acid pre-treatment and a high i.n. dose. Therefore, the findings presented may not represent what occurs in swine. Furthermore, several conclusions are not supported by the data and require substantial new experimental support. Thus, major concerns remain that impact the validity of the findings.

Major concerns for the manuscript:

The intranasal infections were done with S. Suis in the stationary phase which has been shown to have less CPS on the cell wall. While this mimics the literature that shows S. Suis to have less CPS in the CNS, the difference in the pathogenesis of a log phase vs. stationary phage intranasal infection would be interesting. Especially because the bacteria is a part of the natural microbial community of swine tonsils, it is curious if the change in growth phase and therefore CPS levels may be a causative reason for pathogenic invasion in some pigs. To take this line of thought a step further, the authors should consider taking the bacteria from NALT/CSF and blood and compare the lag times bacteria from different organs take to enter a log growth phase to show whether the difference in CPS is because S. Suis in each location is in a different growth phase. If log phase bacteria were intranasally delivered, would it adapt a stationary phase life strategy? How long would that take? Lastly, the authors should be cautious about claims about S. suis downregulating CPS in the NALT for increased invasion and upregulating CPS to survive phagocytosis in blood. While it is true that the data shows that there are different levels of CPS in these locations, the regulation and mechanism of the recorded and observed cell wall difference is not investigated past the correlation to the growth phase. While mechanistic work is outside the scope of the current work, readers should keep in mind that these results may be explained multiple ways. In addition, the mouse model is used rather than the usual host of a pig. The NALTs of conventional pigs and SPF mice certainly have unique microbial communities and this may affect the pathogenesis of S. suis in the mouse, therefore influencing the results. Because the authors show a higher infection rate in the mouse with acetic acid, they may want to consider investigating what the mouse NALT microenvironment is naturally doing to exclude more bacterial invasion in future studies. Is it simply a host mismatch or is there something about the microbiome or steady-state immune system in the nose of mice that is different from pigs?

---

## [Author Response]

The following is the authors’ response to the original reviews.

**Public Reviews:**

**Reviewer 1:**
(1) Some conclusions are not completely supported by the present data, and at times the manuscript is disjoint and hard to follow. While the work has some interesting observations, additional experiments and controls are warranted to support the claims of the manuscript.

Thank you for the comments. We revised some of the claims and conclusions to be more objective and result-supportive.

(2) While the authors present compelling data that is relevant to the development of anti-bacterial vaccinations, the data does not completely match their assertions and there are places where some further investigation would further the impact of their interesting study.

We do not fully agree with the reviewer's comments. We have demonstrated that changes in CPS levels during infection are associated with pathogenesis, which will guide future studies on the underlying mechanisms. A significant amount of effort is required for studying mechanisms, which is beyond the scope of this research. We concur with the reviewer that assertions should be made cautiously until further studies are conducted. We have revised these assertions to align with the data and to avoid extrapolating the results (pages 7, lines 126, 133-136; page 11, lines 216-218; page 13, line 264; and page 18, lines 378-383).

(3) The difference in the pathogenesis of a log phase vs. stationary phage intranasal infection would be interesting. Especially because the bacteria is a part of the natural microbial community of swine tonsils, it is curious if the change in growth phase and therefore CPS levels may be a causative reason for pathogenic invasion in some pigs.

*S. suis* is a part of the natural microbial community of swine tonsils but not mouse NALT. It is interesting to know if CPS levels are low in pig tonsils since CPS is hydrophilic and not conducive to bacterial adhesion. In the study, mice were i.n. infected with a high dose of the bacteria, which could increase opportunities for dissemination (acidic acid may not be a contributor since with or without it is similar). *S. suis* getting into other body compartments from pig tonsils might be triggered by other conditions, such as viral coinfection, nasal cavity inflammation, cold weather, and decreased immunity.

Experiments with pig blood and phagocytes have shown that genes involved in the synthesis of CPS are upregulated in pig blood. In contrast, these genes are downregulated [1]. In addition, the absence of CPS correlated with increased hydrophobicity and phagocytosis, proposing that *S. suis* undergoes CPS phase variation and could play a role in the different steps of *S. suis* infection [2]. We showed direct evidence of encapsulation modulation associated with *S. suis* pathogenesis in mice. A pig infection model is required to confirm these findings.

(4) The authors should consider taking the bacteria from NALT/CSF and blood and compare the lag times bacteria from different organs take to enter a log growth phase to show whether the difference in CPS is because S. suis in each location is in a different growth phase. If log phase bacteria were intranasally delivered, would it adapt a stationary phase life strategy? How long would that take?

What causes CPS regulation in vivo is not known. CPS changes in different culture stages, indicating that stress, such as nutrition levels, is one of the signals triggering CPS regulation. The microenvironment in the body compartments is far more complex than in vitro, in which host cells, immune factors and others may affect CPS regulation, individually or collectively. The reviewer’ question is important but the suggested experiment is impracticable since bacterial numbers taken from organs are few, and culturing the bacteria in vitro would obliterate the in vivo status.

(5) Authors should be cautious about claims about S. suis downregulating CPS in the NALT for increased invasion and upregulating CPS to survive phagocytosis in blood. While it is true that the data shows that there are different levels of CPS in these locations, the regulation and mechanism of the recorded and observed cell wall difference are not investigated past the correlation to the growth phase.

We lower the tone and change the claim as “suggest a correlation between lower CPS in the NALT and a greater capacity for cellular association, whereas elevated CPS levels in the blood are linked to improved resistance against bactericidal activity. However, the mechanisms behind these associations remain unknown.” (page 7, lines 133-136).

(6) The mouse model used in this manuscript is useful but cannot reproduce the nasal environment of the natural pig host. It is not clear if the NALTs of pigs and mice have similar microbial communities and how this may affect the pathogenesis of S. Suis in the mouse. Because the authors show a higher infection rate in the mouse with acetic acid, they may want to consider investigating what the mouse NALT microenvironment is naturally doing to exclude more bacterial invasion. Is it simply a host mismatch or is there something about the microbiome or steady-state immune system in the nose of mice that is different from pigs?

It is a very interesting comment. The mice are SPF level. The microenvironment in SPF mouse NALT should be significantly different from conventional pig tonsils. Although NALT in mice resembles pig tonsils in function, many factors may contribute to the sensitivity to *S. suis* colonization in the pig nasal cavity, such as the microbiome and local steady-state immune system. More complex microbiota in tonsils could be one of the factors. Analyzing what makes *S. suis* inclined towards colonization in pig tonsils by SPF and conventional pigs are an ideal experiment to answer the question.

(7) Have some concerns regarding the images shown for neuroinvasion because I think the authors mistake several compartments of the mouse nasal cavity as well as the olfactory bulb. These issues are critical because neuroinvasion is one of the major conclusions of this work.

Thank you for your comments. The olfactory epithelium (OE) is located directly underneath the olfactory bulb in the olfactory mucosa area and lines approximately half of the nasal cavities of the nasal cavity. The remaining surface of the nasal cavity is lined by respiratory epithelium, which lacks neurons. The olfactory receptor neuron in OE is stained green in the images by β-tubulin III, a neuron-specific marker. The respiratory epithelium is colorless due to the absence of nerve cells. Similarly, the green color stained by β-tubulin III identifies the olfactory bulb. The accuracy of the anatomic compartments of the mouse nasal cavity has been checked and confirmed by referring to related literature [3, 4].

References

(1) Wu Z, Wu C, Shao J, Zhu Z, Wang W, Zhang W, Tang M, Pei N, Fan H, Li J, Yao H, Gu H, Xu X, Lu C. The Streptococcus suis transcriptional landscape reveals adaptation mechanisms in pig blood and cerebrospinal fluid. RNA. 2014 Jun;20(6):882-98.

(2) Charland N, Harel J, Kobisch M, Lacasse S, Gottschalk M. Streptococcus suis serotype 2 mutants deficient in capsular expression. Microbiology (Reading). 1998 Feb;144 (Pt 2):325-332.

(3) Pägelow D, Chhatbar C, Beineke A, Liu X, Nerlich A, van Vorst K, Rohde M, Kalinke U, Förster R, Halle S, Valentin-Weigand P, Hornef MW, Fulde M. The olfactory epithelium as a port of entry in neonatal neurolisteriosis. Nat Commun. 2018;9(1):4269.

(4) Sjölinder H, Jonsson AB. Olfactory nerve--a novel invasion route of Neisseria meningitidis to reach the meninges. PLoS One. 2010 Nov 18;5(11):e14034.

**Reviewer 2:**
(1) However, there are serious concerns about data collection and interpretation that require further data to provide an accurate conclusion. Some of these concerns are highlighted below:

Both reviewers were concerned about some of the interpretations of the results. We modified the interpretations in related lines throughout the manuscript (Please see the related responses to Reviewer 1).

(2) In figure 2, the authors conclude that high levels of CPS confer resistance to phagocytic killing in blood exposed S. suis. However, it seems equally likely that this is resistance against complement mediated killing. It would be important to compare S. suis killing in animals depleted of complement components (C3 and C5-9).

We thank the reviewer for the comment. The experiment should be Bactericidal Assay instead of anti-phagocytosis killing. CPS is a main inhibitor of C3b deposition [1]. It interferes with complement-mediated and receptor-mediated phagocytosis; and direct killing. Data in Figure 2C is expressed as “% of bacterial survival in whole blood” for clarity (page 8, Fig. 2C and page 23, lines 489-490).

(3) Intranasal administration non-CPS antisera provides a nice contrast to intravenous administration, especially in light of the recently identified "blood-olfactory barrier". Can the authors provide any insight into how long and where this antibody would be located after intranasal administration? Would this be antibody mediated cellular resistance, or something akin to simple antibody "neutralization"

Anti-V5 may not stay long locally following intranasal administration. Efficient reduction of *S. suis* colonization in NALT supports that anti-V5 could recognize and neutralize the bacteria in NALT quickly, thereby reducing further dissemination in the body. Antibody-mediated phagocytosis may not play a major role because neutrophils are mainly present in the blood but not in the tissues.

(4) The micrographs in Figure 7 depict anatomy from the respiratory mucosa. While there is no histochemical identification of neurons, the tissues labeled OE are almost certainly not olfactory and in fact respiratory. However, more troubling is that in figures 7A,a,b,e, and f, the lateral nasal organ has been labeled as the olfactory bulb. This undermines the conclusion of CNS invasion, and also draws into question other experiments in which the brain and CSF are measured.

We understand the significance of your concerns and appreciate your careful review of Figure 7. The olfactory epithelium (OE) is situated directly beneath the olfactory bulb in the olfactory mucosa area and covers about half of the nasal cavity. This positioning allows information transduction between the olfactory and the olfactory epithelium. The remaining surface of the nasal cavity is lined with respiratory epithelium, which does not contain neurons and primarily serves as a protective barrier. In contrast, the olfactory epithelium consists of basal cells, sustentacular cells, and olfactory receptor neurons. The olfactory receptor neurons are specifically stained green in the images using β-tubulin III, a marker that is unique to neurons. The respiratory epithelium appears colorless due to the lack of nerve cells. Similarly, the green staining with β-tubulin III also highlights the olfactory bulb. The anatomical structures indicated in the images are consistent with those described in the literature [2, 3], confirming that the anatomy of the nasal cavity has been accurately identified.

(5) Micrographs of brain tissue in 7B are taken from distal parts of the brain, whereas if olfactory neuroinvasion were occurring, the bacteria would be expected to arrive in the olfactory bulb. It's also difficult to understand how an inflammatory process would be developed to this point in the brain -even if we were looking at the appropriate region of the brain -within an hour of inoculation (is there a control for acetic acid induced brain inflammation?). Some explanations about the speed of the immune responses recorded are warranted.

Thank you for highlighting this issue. Cerebrospinal fluid (CSF) flows into the subarachnoid space surrounding the spinal cord and the brain. There are direct connections from this subarachnoid space to lymphatic vessels that wrap around the olfactory nerves as they cross the cribriform plate towards the nasal submucosa. This connection allows for the drainage of CSF into the nasal submucosal lymphatics in mice [4, 5]. Bacteria may utilize this CSF outflow channel in the opposite direction, which explains the development of brain inflammation in the distal areas of brain tissue adjacent to the subarachnoid space. We have included additional relevant information in the revised manuscript (page 16, lines 323-325).

(6) The detected presence of S. suis in the CSF 0.5hr following intranasal inoculation is difficult to understand from an anatomical perspective. This is especially true when the amount of S. suis is nearly the same as that found within the NALT. Even motile pathogens would need far longer than 0.5hr to get into the brain, so it's exceedingly difficult to understand how this could occur so extensively in under an hour. The authors are quantifying CSF as anything that comes out of the brain after mincing. Firstly, this should more accurately be referred to as "brain", not CSF. Secondly, is it possible that the lateral nasal organ -which is mistakenly identified as olfactory bulb in figure 7- is being included in the CNS processing? This would explain the equivalent amounts of S. suis in NALT and "CSF".

The high dose of inoculation used in the experiment may explain the rapid presence of *S. suis* in the CSF. Mice exhibit low sensitivity to *S. suis* infection, and the range for the effective intranasal infectious dose is quite narrow. Higher doses lead to the quick death of the mice, while lower doses do not initiate an infection at all. The dose used in this study is empirical and is intended to facilitate the observation of the progression of *S. suis* infection in mice.

The NALT tissue and CSF samples are collected separately. After obtaining the NALT tissue, the nasal portion was carefully separated from the rest of the head along the line of the eyeballs. The brain tissue was then extracted from the remaining part of the head to collect the CSF, and it was lacerated to expose the subarachnoid space without being minced. This procedure aims to preserve the integrity of the brain tissue as much as possible. Further details about the CSF collection process can be found in the Materials and Methods section (page 24, lines 508-512).

(7) To support their conclusions about neuroinvasion along the olfactory route and /CSF titer the authors should provide more compelling images to support this conclusion: sections stained for neurons and S. suis, images of the actual olfactory bulb (neurons, glomerular structure etc).

Thank you. We respectfully disagree with the reviewer. We stained neurons using a neuron-specific marker to identify the anatomical structures of the olfactory bulb and olfactory epithelium (in green). We used an *S. suis*-specific antibody to highlight the bacteria present in these areas (in orange and red). The images, along with the bacteria found in the cerebrospinal fluid (CSF) and the brain inflammation observed early in the infection, strongly support our conclusion regarding brain invasion through the olfactory pathway. Please see the response to question 4 for further clarification.

References

(1) Seitz M, Beineke A, Singpiel A, Willenborg J, Dutow P, Goethe R, Valentin-Weigand P, Klos A, Baums CG. Role of capsule and suilysin in mucosal infection of complement-deficient mice with Streptococcus suis. Infect Immun. 2014 Jun;82(6):2460-71.

(2) Sjölinder H, Jonsson AB. Olfactory nerve--a novel invasion route of Neisseria meningitidis to reach the meninges. PLoS One. 2010 Nov 18;5(11):e14034.

(3) Pägelow D, Chhatbar C, Beineke A, Liu X, Nerlich A, van Vorst K, Rohde M, Kalinke U, Förster R, Halle S, Valentin-Weigand P, Hornef MW, Fulde M. The olfactory epithelium as a port of entry in neonatal neurolisteriosis. Nat Commun. 2018;9(1):4269.

(4) Yoon JH, Jin H, Kim HJ, Hong SP, Yang MJ, Ahn JH, Kim YC, Seo J, Lee Y, McDonald DM, Davis MJ, Koh GY. Nasopharyngeal lymphatic plexus is a hub for cerebrospinal fluid drainage. Nature. 2024 Jan;625(7996):768-777.

(5) Spera I, Cousin N, Ries M, Kedracka A, Castillo A, Aleandri S, Vladymyrov M, Mapunda JA, Engelhardt B, Luciani P, Detmar M, Proulx ST. Open pathways for cerebrospinal fluid outflow at the cribriform plate along the olfactory nerves. EBioMedicine. 2023 May;91:104558.

**Response to Recommendations for the authors:**

**Reviewer 1:**
Minor concerns for the manuscript:(1) In the introduction, please consider giving a little more background about the bacteria itself and how it causes pathogenesis.

We appreciate your suggestion. We have included additional background on the virulent factors and the pathogenesis of the bacteria in the introduction to enhance understanding of the results (page 4, lines 63-69).

(2) Figure 2C would be more correct to say percent survival as the CFUs before and after are what are being compared and not if the bacteria is being phagocytosed or not. Flow cytometry of the leukocytes and a fluorescent S. Suis would show phagocytosis. Unless that experiment is performed, the authors cannot claim that there is a resistance to phagocytosis.

Thank you for your feedback. We agree with the reviewer that the experiment should be Bactericidal Assay rather than anti-phagocytosis killing. CPS interferes with complement-mediated phagocytosis and direct killing, and receptor-mediated phagocytosis. To enhance clarity, the data in Fig. 2C has been presented as “% of bacterial survival in whole blood” (page 8).

(3) There are two different legends present for Figure 1. Please resolve.

We apologize for the oversight. The redundant figure legend has been removed (page 6).

(4) There are places such as in lines 194-195, that there are assertions and interpretations about the data that are not directly drawn from the data. These hypotheses are valuable, but please move them to the discussion.

Thank you for your suggestion. The hypothesis has been moved to the Discussion section (page 19, lines 402 - 405).

(5) In Figure 4B, higher resolution images would strengthen the ability of non-microbiologists to see the differences in CPS levels in the cell wall.

We achieved the highest resolution possible for clearer distinctions in CPS levels. To enhance the visualization of the different CPS levels in the images, we revised the description of the CPS changes in Figure 4B within the results section (page 11, lines 208-213).

(6) In Figure 5 there is no D. Further, the schematics throughout would be easier to parse with the text if the challenge occurred at time 0. Consider revising them for clarity.

Thank you for highlighting the error. We have removed "i.v + i.n (Fig. 5)" from Figure 5A and made adjustments to the schematic illustrations in Figures 5 and 6 as recommended by the reviewer (page 14).

(7) What is the control for the serum? The findings for figures 5 and 6 would be much stronger if a non- S. Suis isotype control serum was also infused.

We used a naive serum as a control to avoid interference from a non-*S. suis* isotype control that targets other surface molecules of *S. suis* serotypes.

(8) Figure 6 legend does not include the anti-CPS treatment.

Thank you. We have added anti-CPS serum in the legend (page 15, line 249).

(9) Figure 7 legend does not include the time point for panel 7A.

Thank you. The time point is shown on Fig.7A (page 17).

(10) Figure 7 should show OB micrographs or entire brain including the OB.

The neuron-specific marker, β-tubulin III, identifies the neuro cells in the olfactory bulb (OB) as shown in Fig. 7A. Unfortunately, we were unable to provide an image of the entire brain that includes the OB due to limitations in our section preparation. We apologize for the mislabeled structure in Fig. 7A, which may have caused confusion. We have corrected the labeling for consistency (see page 15, lines 257-260). Additionally, we included a drawing of the sagittal plane of the rodent's nose, depicting the compartments of the OB, olfactory epithelium (OE), nasal cavity (NC), and brain. This illustration, presented in Fig. 7B on page 17, aims to clarify the structural and functional connections between the nasopharynx and the CNS.

(11) Some conclusions may be better drawn if figures were to be consolidated. As noted above, the data at times feels disjointed and the importance is more difficult for readers to follow because data are presented further apart. Particularly figures 5 and 6 which are similar with different time points and controls of antisera administrative routes; placing these figures together would be an example of increasing continuity throughout the paper.

Thank you for the valuable suggestion. Figures 5 and 6, along with their related descriptions in the results section, have been combined for better cohesiveness (pages 14-15).

**Reviewer #2:**
To support their conclusions about neuroinvasion along the olfactory route and /CSF titer the authors should provide more compelling images to support this conclusion: sections stained for neurons and S. suis, images of the actual olfactory bulb (neurons, glomerular structure etc).

Please refer to our responses to Reviewer 1's Question 7, Reviewer 2's Questions 4 and 7 in the public reviews, and Reviewer 1's Question 10 in the authors' recommendations.